# The Role of RNA Modification in HIV-1 Infection

**DOI:** 10.3390/ijms23147571

**Published:** 2022-07-08

**Authors:** Shuqi Wang, Huanxiang Li, Zhengxing Lian, Shoulong Deng

**Affiliations:** 1Beijing Key Laboratory for Animal Genetic Improvement, National Engineering Laboratory for Animal Breeding, Key Laboratory of Animal Genetics and Breeding of the Ministry of Agriculture, College of Animal Science and Technology, China Agricultural University, Beijing 100193, China; s20213040591@cau.edu.cn (S.W.); sy20213040735@cau.edu.cn (H.L.); 2NHC Key Laboratory of Human Disease Comparative Medicine, Institute of Laboratory Animal Sciences, Chinese Academy of Medical Sciences and Comparative Medicine Center, Peking Union Medical College, Beijing 100021, China

**Keywords:** RNA modification, HIV-1 infection, RNA methylation modification, RNA virus, post-transcriptional modification

## Abstract

RNA plays an important role in biology, and more than 170 RNA modifications have been identified so far. Post-transcriptional modification of RNA in cells plays a crucial role in the regulation of its stability, transport, processing, and gene expression. So far, the research on RNA modification and the exact role of its enzymes is becoming more and more comprehensive. Human immunodeficiency virus 1 (HIV-1) is an RNA virus and the causative agent of acquired immunodeficiency syndrome (AIDS), which is one of the most devastating viral pandemics in history. More and more studies have shown that HIV has RNA modifications and regulation of its gene expression during infection and replication. This review focuses on several RNA modifications and their regulatory roles as well as the roles that different RNA modifications play during HIV-1 infection, in order to find new approaches for the development of anti-HIV-1 therapeutics.

## 1. Introduction

Post-transcriptional modification of RNA is a common phenomenon in biological processes [1]. Since the discovery of the first structurally modified nucleoside pseudouridine in the 1950s, more than 170 types of RNA modifications have been identified in cells, including messenger RNA (mRNA), transfer RNA (tRNA), ribosomal RNA (rRNA), small non-coding RNAs, and long non-coding RNAs (lncRNA) [2,3,4]. Similar to epigenetic DNA methylation, epigenetic modifications are regulated by different types of regulators and can be written, read, and cleared by complex protein networks [4]. Furthermore, RNA modifications at the cellular level regulate a variety of cellular processes, including cell death, proliferation, aging, differentiation, migration, metabolism, autophagy, DNA damage response, etc. They play an important role in the physiology and etiology of many diseases [5].

RNA methylation is the most well-characterized modification of RNA modifications. According to the site of methylation, RNA methylation can be divided into various forms, including *N*1-methyladenosine (m^1^A), 5-methylcytosine (m^5^C), *N*6-methyladenosine (m^6^A), *N*7-methylguanosine (m^7^G) and *N*6,2′-*O*-dimethyladenosine (m^6^Am), etc. [3,6]. The highly dynamic and reversible methylation of RNA is mediated by RNA-modification proteins (RMPs). RPMs include “writers”, “erasers” and “readers”. “Writers”” write” methylation modifications into RNA, which mediates the methylation modification process of RNA. “Erasers” “erase” RNA methylation signals, which mediates demethylation modification of RNA. “Readers” “read” the information of RNA modification, and participate in downstream RNA translation, degradation and other processes [7].

Although our understanding of the importance of RNA modification in physiological and pathological processes is rapidly increasing, little is known about the function of RNA modification in the mammalian immune system or its role in host-pathogen interactions. In this paper, the m^6^A, m^1^A, m^5^C, m^7^G, m^6^Am, pseudouridine (ψ), 2′-*O*-methylation (2′-*O*-Me or Nm) and *N*4-acetylcytidine (ac^4^C) modifications (Figure 1) as well as their roles in HIV-1 infection are introduced.

## 2. RNA Modifications

### 2.1. m^6^A

m^6^A is the most prevalent RNA modification in many species including mammals and is present in the 5′UTR, 3′UTR and stop codons [8,9,10]. m^6^A modifications act as epitope transcriptional markers to regulate almost all aspects of RNA transcript metabolism, including splicing, stability, structure, translation, and export [11,12,13,14,15].

Although m^6^A was discovered in 1974, thanks to the improvement of analytical chemistry and high-throughput sequencing detection methods in recent years, great progress has been made in the study of mRNA modification and its physiological importance [16]. First, the determination of fat mass and obesity-related protein (FTO) and the demethylase ALKBH5 provided evidence for the reversible regulation of RNA modifications [17,18]; In addition, the use of high-throughput sequencing has provided a transcriptome-wide map of modification sites in mRNA and lncRNA and revealed that m^6^A sites are enriched near stop codons and in the 3′UTR [9]. These two advances have greatly promoted the study of internal mRNA modification functions [3].

The m^6^A modification is catalyzed by a stable heterodimeric core complex formed by methyltransferase-like 3 (METTL3) and methyltransferase-like 14 (METTL14). METTL3 is the is a functional active site as a catalytic subunit, and METTL14 is an essential site that promotes RNA binding and is a scaffolding structure [19,20]. Further studies found that Wilms Tumor 1 Association Protein binds to METTL3/14 and is required for optimal substrate recruitment and METTL3/14 localization [3,19,20,21,22,23]. In addition, vir-like m^6^A methyltransferase-related (VIRMA; also known as KIAA1429) [24], METTL16 [25], RNA binding motif protein 15 (RBM15) [26], zinc finger CCCH-type containing protein 13 (Zc3h13) [27] and the cbl proto-oncogene-like protein 1 (CBLL1, also known as HAKAI) [28] are essential for the nuclear localization and stabilization of the “writer” complex, or the deposition specificity of m^6^A [29]. The “erasers” of m^6^A includes the fat mass and obesity-related protein (FTO) and the alkylated DNA repair protein ALKB5 (ALKBH5), which are mainly present in the cytoplasm or nucleus, respectively, and can selectively demethylate RNA [30,31,32]. FTO not only functions as a demethylase of m^6^A in mRNA, but also shows demethylase activity of m^6^A m and m^1^A of specific tRNA, snRNA or mRNA [30]. “Readers” that decode m^6^A methylation and generate functional signals include the YTH-RNA-binding domain family (YTHDF1-3, YTDHC1-2) [33,34], the heterogeneous nuclear protein (HNRNP) family (HNRNPA2B1, HNRNPC, HNRNPG) [35,36], IGF2 mRNA binding proteins (IGF2BPs) [37], eukaryotic initiation factor (eIF) 3 [38], HUR [39], fragile X mental retardation protein (FMRP) [40] and leucine-rich pentatricopeptide repeat-containing (LRPPRC) [41]. All three proteins of YTHDF1-3 contain a conserved carboxy-terminal m^6^A-binding YTH domain and a variable N-terminal effector domain with unclear function [42]. Although “writers” and “erasers” can write or remove RNA modifications, the role of “readers” is essential for the regulation of gene expression by RNA modifications. The m^6^A modification adds a new layer of post-transcriptional gene expression regulation that is involved in T cell response to HIV infection, production of type I interferons, and T cell differentiation and homeostasis (Table 1).

### 2.2. m^1^A

m^1^A is a reversible modification of mRNA and tRNA and is a positively electrostatically methylated nucleotide under physiological conditions [83]. Methylation of adenosine N1 occurs at the Watson-Crick interface, blocking base pairing, thereby affecting reverse transcription and protein translation [84]. The content of m^1^A is high in tRNA and rRNA but low in mRNA. The mRNA containing m^1^A is 10 times less than that containing m^6^A [84,85]. m^1^A was first identified in tRNAs, usually located at positions 9, 14, and 58 of tRNAs [86]. m^1^A is critical for tRNA folding, stability, and tRNA-protein interactions [87,88]. m^1^A is prevalent in mitochondrial-encoded transcripts, and through the study of manipulating m^1^A level by a mitochondrial located m^1^A methyltransferase trmt61b, it was found that m^1^A interferes with the translation of mitochondrial mRNA [43]. Methylation of tRNA is accomplished by a family of RNA methyltransferases (mases), which has more than 60 members in humans [89]. TRMT61A and TRMT6 are responsible for m^1^A58 modification of cytoplasmic tRNAs [44]. In addition, TRMT10C and TRMT61B catalyze m^1^A at positions 9 and 58 in mitochondrial tRNA [45,46]; in human 28S rRNA, m^1^A is methylated at position 1309 by the nucleolar protein RRP8 (also known as NML), and this methylation is required for rRNA biogenesis [90,91]. In mRNA, TRMT6/61A recognizes the T-loop-like structure of the GUUCRA tRNA-like motif therein and modifies it with m^1^A, TRMT61B installs m^1^A in mt-mRNA transcripts, and TRMT10C methylates position 1374 of ND5 mt-mRNA [43,47]. m^1^A can be catalyzed by the demethylases FTO, ALKBH1 and ALKBH3 [30,48,49,85]. YTHDF1, YTHDF2, YTHDF3 and YTHDC1 have been reported as “readers” by binding to RNAs with m^1^A [50,51].

### 2.3. m^5^C

The discovery of 5-methylcytosine (m^5^C) of DNA dates back to the 1950s and is essential for gene expression and epigenetic regulation [92,93]. In the 1970s, researchers discovered the existence of m^5^C in RNA, and high-throughput detection methods found that m^5^C is an RNA modification widely present in mRNA and lncRNA, including transfer RNA (tRNA), ribosomal RNA (rRNA), Long non-coding RNA (lncRNA), small nuclear RNA (snRNA), microRNA (miRNA), and enhancer RNA (eRNA) [94]. Modifications at specific positions of m^5^C in mRNA show different regulatory activities, they can promote or inhibit translation; in tRNA, m^5^C can stabilize secondary structure and affect anticodon stem-loop conformation; in rRNA, m^5^C can affect translation preservation Authenticity [52]. The m^5^C methyltransferase usually uses S-adenosylmethionine (SAM) as the methyl donor to transfer the methyl group to the fifth carbon position of the RNA cytosine base [95]. More than 10 RNA m^5^C methyltransferases have been identified, including enzymes of the NOL1/NOP2/SUN domain (NSUN) family, DNA methyltransferase (DNMT) homolog DNMT2, and tRNA specific methyltransferase (TRDMT) family members [53]. The erasers of m^5^C are somewhat controversial and have yet to be determined. In recent years, some reports indicate that the ten-eleven translocator (Tet) family has the potential to act as an RNA demethylase. In mRNA, enzymes of TET can oxidize m^5^C to generate 5-hydroxymethylcytosine (hm^5^C) [96]. Furthermore, Alpha-ketoglutarate-dependent dioxygenase ABH1(ALKBH1) catalyzes m^5^C34 to f^5^Cm34 (5-formyl-2′-*O*-methylcytidine) and hm^5^Cm34 (5-hydroxymethyl-2′-*O*-methylcytidine) along mt-tRNA^Met^ and anticodon cytoplasmic tRNA^Leu^ [97,98]. RNA m^5^C reader proteins ALYREF (RNA and export factor-binding protein 2) and YBX1 (Y-box-binding protein 1) exert biological effects by recognizing and binding m^5^C sites [54,55,56].

### 2.4. m^7^G

m^7^G, one of the ubiquitous RNA modifications in various species, is a positively charged RNA modification produced by the addition of a methyl group to the N7 position of nucleoguanosine [99,100]. Eukaryotic mRNAs are capped with 7-methylguanosine nucleotides at their 5′ ends, a structural feature that enhances mRNA stability, is critical for mRNA splicing, export to cytoplasm, and efficient translation into proteins, and increases translation efficiency [57]. In addition, m^7^G also regulates miRNA biosynthesis and biological function, tRNA stability, 18S rRNA processing and maturation [59]. m^7^G -containing tRNAs are present in eukaryotes, they are primarily found in yeast and more recently in mice, and this modification is catalyzed by the Trm8/Trm82 complex in yeast and the METTL1/WDR4 complex in humans [58]. Studies have shown that WBSCR22 is required for G1639 N(7) methylation in 18S rRNA in humans [101], but its role is not fully understood.

### 2.5. m^6^Am

An abundant RNA modification close to the mRNA cap structure is dimethylated adenosine, m^6^Am [60]. The first nucleotide after the m^7^G cap can be methylated on the ribose to form 2′-O-methyladenosine (Am), which can then be further methylated at its N6 position to generate m^6^Am [57,102]. m^6^Am is the most prevalent RNA modification in many species, including mammals, present in the 5′-UTR, 3′-UTR and stop codons, and in the first transcription nucleus in about 30% of cellular mRNAs nucleotides can have a significant impact on gene expression in the transcriptome [60]. Recent studies have found that phosphorylated C-terminal domain-interacting factor 1 (PCIF1) is the enzyme that catalyzes m^6^Am [61,62].

### 2.6. *Ψ*

Ψ, the C5 glycoside isomer of guanosine, was the first post-transcriptional modification discovered, the fifth nucleotide of RNA, and the most abundant modification in RNA [103]. Ψ is widely present in stable ncRNAs, including rRNA, tRNA and snRNA [64]. Among these abundant ncRNAs, Ψ plays an important role in regulating their functions. For example, Ψ is required for rRNA to properly fold and ensure its translational fidelity [63]; in tRNA, Ψ can stabilize its structure; in snRNA, Ψ can affect snRNP biogenesis and mRNA splicing [64,65]. Pseudouridylation can be achieved by two different mechanisms, RNA-independent pseudouridylation and RNA-dependent pseudouridylation. The RNA-dependent mechanism relies on RNA-protein complexes called Box H/ACA snoRNA small ribonucleoproteins (snoRNPs), which consist of a boxH/ACA snoRNA and four core proteins: dyskerin (also known as NAP57 or DKC1), non-histone protein2 (Nhp2), nucleolar protein 10 (Nop10) and glycine-arginini-rich prtein1 (Gar1). In H/ACA, ncRNAs are responsible for substrates through complementary base-pairing interactions with RNA substrates recognition, catalytic activity is provided by DKC1 [66]. RNA-independent pseudouridylation is catalyzed by pseudouridine synthase (PUS), which enables simultaneous substrate recognition and catalysis in the absence of RNA template strands [67]. In eukaryotes, there are more than 14 different PUS enzymes, 10 of which belong to the PUS family, including Pus1-10 [68], each of them has specific substrates but also share some substrates.

### 2.7. Nm

2′-*O*-methylation (2′-*O*-Me or Nm, where N represents any nucleotide) is an abundant and highly conserved modification. Nm is modified to add a methyl group to the 2′ hydroxyl group of the ribose moiety, any of the four nucleotides (A, C, U, or G) may have ribose methyl groups at different stages of the ribosome biogenesis process. In addition, Nm is abundant in tRNA, rRNA and SnRNA, which is important for the biogenesis, metabolism, and function of these molecules [69,70,71]. Nm has also been found in mRNAs and SNcRNAs, such as plant microRNAs (miRNA) and small interfering RNAs (siRNA) [104,105,106]. Methylation of ribose increases the hydrophobicity of RNA molecules and protects the RNA backbone from nuclear cleavage attacks, stabilizes RNA structure, and can affect potential interactions with proteins or other RNAs [72,107]. The 2′-*O*-methylation of ribose in RNA is formed by two cellular mechanisms: through independent methyltransferases or through complex assembly of proteins associated with snoRNA guides (snoRNPs) [108]. Most mRNAs and miRNAs, some tRNAs, and all bacterial rRNAs and tRNAs are modified by independent methyltransferases [109]. The human tRNA 2′-*O*-methyltransferase FTSJ1, also known as MRX9, TRMT7, JM23, can target the C32 and N34 positions in the anticodons of tRNA^Phe^ and tRNA^Trp^ [73]. FTSJ1 has two conserved analogous sequences, FTSJ2 (Saccharomyces cerevisiae mitochondrial rRNA methyltransferase 2 [MRM2]) for 21S rRNA and STSJ3 (Saccharomyces cerevisiae poly A-binding protein Pab1p) for 25S rRNA [74,75]. In eukaryotic ribosomes, ribose methylation is carried out by the methyltransferase Fibrillarin (FBL), part of the ribonucleoprotein complex, which is guided to its targets by box C/D snoRNAs [76]. The G2922 position on yeast 25S rRNA is modified by the 2′-*O*-methyltransferase Sbp1 (human homolog: FTSJ3), which does not involve an antisense guide RNA [77].

### 2.8. ac^4^C

ac^4^C is an ancient and conserved RNA modification that was first described in the *E. coli* tRNA met anticodon [110,111], and subsequently in budding yeast and human tRNA, serine and leucine tRNA as well as 18S rRNA [112,113]. ac^4^C is widely distributed in the human transcriptome, with most sites occurring in the coding sequence (CDS), and acetylation can promote the expression of target genes by improving mRNA stability and translation [78]. All the characteristic sites of ac^4^C in eukaryotic RNA occur in the central nucleotide of the 5′-CCG-3′ consensus sequence [114]. For tRNA, the ac^4^C modification contributes to the correct recognition of codons, stabilizes the tertiary structure of the tRNA molecule, increases the high fidelity of protein translation, and plays a key role in cellular thermotolerance [79,80]; The ac^4^C modification on rRNA is a characteristic of thermophilic organisms [115], it also plays an important role in the accurate translation of proteins [81]. In addition, ac^4^C modification can also increase mRNA stability, enhance substrate translation, and promote mRNA translation efficiency [78]. *N*-acetyltransferase 10 (NAT10) is involved in catalyzing the deposition of ac^4^C on tRNA, rRNA and mRNA, and acetyl coenzyme A (CoA) acts as an acetyl donor [78]. Furthermore, the formation of ac^4^C catalyzed by NAT10 on tRNA and rRNA requires the assistance of the cofactors THUMPD1 and snoRNA, respectively [82]. Whether various RNAs can be deacetylated is currently unclear and requires further research.

## 3. The Role of RNA Modifications in HIV-1 Infection

Human Immunodeficiency Virus (HIV) is a virus that attacks the body’s immune system. HIV was first isolated and identified in 1983 from the lymph nodes of a patient with lymphadenopathy [116,117]. The disease caused by HIV infection is called acquired immunodeficiency syndrome, also known as AIDS. People with this disease partially or completely lose their immune function, reduce the number of CD4+ cells, and then develop opportunistic infections, tumors, etc., with various clinical manifestations [118]. AIDS spreads rapidly, has a high case-fatality rate, and is currently incurable, except for drugs to delay death [119].

In terms of virus taxonomy, HIV belongs to the genus Lentivirus of the Retroviridae family, and is currently classified into type 1 and type 2 (HIV-1, HIV-2) according to their genetic characteristics and differences in viral antigens [120]. Compared with HIV-2, HIV-1 has a shorter incubation period, higher incidence, and higher transmission rate. HIV-2 is less virulent and transmissible than HIV-1, causes a slower and milder course of AIDS, and has a lower mortality rate than type 1 [121]. The current research is also mainly carried out on HIV-1.

The HIV genome consists of two identical single-stranded RNA molecules enclosed within the core of the viral particle. The HIV genome, also called previral DNA, is produced by reverse transcription of viral RNA genomic RNA into DNA, degradation of RNA, and integration of double-stranded HIV DNA into the human genome [120].

The replication cycle of HIV can be divided into early and late stages. In the early stage, the virus binds to the prime cell surface receptor, then enters the cell, the viral RNA is reverse transcribed into DNA, the viral capsid is uncoated, the viral DNA is imported into the nucleus, and then the DNA is integrated, and the viral DNA is integrated into the host cell genome. Mature HIV-1 surface glycoprotein gp120 binds to CD4 receptors on host cells [122]. After the conformational change occurs, it binds to the co-receptors CCR5/CXCR4 [122,123], to further trigger the conformational change of gp41 [124]. The N-terminus of gp41 exists on the viral membrane, forming a channel and inserting into the host cell plasma membrane [120]. Then the viral envelope and cell membrane are completely fused, and the viral nucleic acid, reverse transcriptase, integrase and protease required for virus replication enter the host cell. HIV-1 reverse transcription transcribes single stranded RNA into DNA, and then uses single stranded DNA as a template to synthesize double stranded DNA under the action of DNA polymerase. Then the double stranded DNA of the virus enters the nucleus and integrates into the DNA strand of the host cell chromosome to form a proviral DNA [125]. Late stages of the replication cycle are transcription of viral genes, export of viral RNA from nucleus to cytoplasm, translation of viral RNA to produce viral structural proteins, assembly of structural proteins with new viral genomes, germination of new virions from infected cells, and particle maturation, which is the process from gene expression to maturation of new virion release [126]. In this process, the provirus is activated for self-transcription, and the viral DNA is used as a template to transcribe and synthesize RNA. Viral RNAs are exported from the nucleus to the cytoplasm, where viral proteins are synthesized. gag polyproteins are assembled on the plasma membrane, and gag interacts with dimeric viral RNA during or after transport to the plasma membrane. Viral envelope proteins accumulate in the plasma membrane, where virus assembly takes place. gag then recruits and hijacks the ESCRT machinery that produces the membrane rupture reaction. During or shortly after the germination of particles from the cell surface. The viral protease lyses the GAG polyprotein precursor, triggering HIV-1 maturation [126].

The HIV-1 genome encodes nine open reading frames, 15 different proteins. Both ends of the genome have long terminal repeats (LTRs) (Figure 2). There are Gag, Pol and Env three reading frames encoding a variety of proteins, The Gag reading frame encodes outer nuclear membrane protein (MA), capsid protein (CA), nucleocapsid protein (NC) and a smaller nucleic acid stabilizing protein p6; the Env reading frame encodes two envelope glycoprotein surface proteins (SU) and transmembrane protein (PR), Env protein and Gag protein constitute the core and outer membrane envelope of virions; the Pol reading frame encodes protease (PR), reverse transcriptase (RT) and integrase (IN), and provides basic enzymatic functions. In addition to structural proteins, HIV encodes six additional proteins, transactivator (Tat) and RNA splicing regulator (Rev) that provide gene regulatory functions and are necessary to initiate HIV replication; Negative regulatory factor (Nef), viral infective factor (Vif) and viral protein r (Vpr) are present in viral particles, virus protein unique (Vpu) indirectly assists the assembly of virus particles, The above four proteins have effects on virus replication, virus germination and pathogenesis [120,127,128]. The export of intron-containing viral RNA is mediated by the Rev-responsive element (RRE), a highly structured cis-acting RNA element [126]. RRE is present in all HIV mRNA sequences and binds Rev (forming a Rev-Rre ribonucleoprotein complex) and various cytokines to overcome cellular defense mechanisms that normally prevent the exit of foreign mRNA molecules [129].

Replication and infection of HIV-1 virus are also regulated by RNA modifications. Once HIV-1 proviral DNA is integrated into the host cell chromosome, viral latency is affected by post-transcriptional modifications (acetylation/deacetylation) [130]. For viral gene expression, the expression program synthesizes 15 proteins from a single transcript of a full-length 9kb mRNA. Production of mechanical proteins and enzymes is dependent on full-length mRNA in unspliced form. HIV-1 gene expression is also involved in other complex post-transcriptional regulation, including non-canonical nuclear export controlled by the viral protein REV and ribosome recruitment using cap-dependent and cap-independent mechanisms. In the process of virus replication, it is promoted by m^6^A, m^5^C RNA modification [2]. The m^6^A modification of HIV-1 RNA plays an important role in viral infection and HIV-1 protein synthesis [131]. The effects of different RNA modifications on HIV-1 infection are described in detail below (Table 2).

### 3.1. The Role of m^6^A Modification during HIV Infection

Internal m^6^A modifications were also found in viral RNA decades ago in Rous sarcoma virus [132], influenza virus [133] and SV40 virus [134]. However, the functional capacity and relevance of this modification to viral replication remains unclear, and whether and how viral infection alters the dynamics of host or viral RNA methylomes remains fundamental and unanswered questions. In the study of Lichinchi et al., it was reported for the first time that m^6^A modification exists in HIV-1 RNA and is abundant in the entire viral genome, present in regulatory regions, coding sequences and structural regions. The results suggest that HIV-1 infection of T cells promotes methylation of viral and host RNAs, and that RNA methylation status is positively correlated with viral replication. Silencing of ALKBH5 increases RRE RNA methylation, which promotes REV binding and increases nuclear export of viral RNA, which promotes viral replication; In contrast, silencing of METTL3 and/or METTL14 reduced RRE RNA methylation and subsequent recruitment of Rev to the RRE, thereby reducing viral RNA export and inhibiting viral replication. In vitro experiments determined that efficient Rev binding requires A7883 in the RRE structure, one that flips out from the RRE stem region and interacts directly with REV’s W45 via van der Waals contacts [135]. Furthermore, mutation of W45 can inactivate Rev [136].

In the study of Kennedy et al., the m^6^A modification sites on the HIV-1 RNA genome were precisely located, mainly located in the 3′ untranslated regions (3′ UTRs) of various HIV-1 mRNAs. Studies have shown that m^6^A residues in the 3′ non-coding region can promote the expression and replication of HIV-1 gene by increasing the stable expression level of viral mRNA. When YTHDF1-3 was overexpressed, HIV-1 gene expression was affected in 293T cells, and increased levels of Gag, Nef, Tat, and Rev mRNAs were observed, as well as increased protein levels of Gag and Nef. Furthermore, HIV-1 is highly sensitive to m^6^A “reader” YTHDF2 expression levels in infected T cells, and the expression level of YTHDF2 is positively correlated with its replication [42].

In the study of Tirumuru et al., it was found that in 293T cells, the overexpression of m^6^A “readers” led to a decrease in the level of HIV-1 infection. In primary CD4+ T cells, HIV-1 RNA is modified by m^6^A in infected cells, and “readers” YTHDF1-3 protein inhibits viral reverse transcription and gene expression by binding to m^6^A -modified HIV-1 RNA, thereby inhibiting HIV-1 infection. Knockdown of m^6^A “writers” reduced HIV-1 Gag synthesis and virion release in virus-producing cells, whereas knockdown of “erasers” had the opposite effect [131].

Regarding the role of m^6^A “writers” and “erasers” in HIV-1 gene expression, the above experiments demonstrated that knockdown of m^6^A “writers” METTL3 and METTL14 resulted in decreased Cap24 and Gag protein levels and total gp120 mRNA levels in cells and supernatants decreased, whereas knockdown of “erasers” FTO and/or had the opposite effect. Although the roles of m^6^A “writers” and “erasers” in HIV-1 gene expression are clear, the role of m^6^A “readers” remains controversial. Using different methods, cell lines, and other reagents may have contributed to different results when studying the relationship between YTHDF protein and HIV-1 replication and expression, so these results need to be clarified by continued research.

A new study shows that in human CD4+ T cells, the YTHDF3 protein is incorporated into HIV particles in a nucleocapsid-dependent manner and is able to reduce viral infectivity during the next infection cycle, while HIV restricts the viral activity of YTHDF3 by cleaving virion-bound YTHDF3 at different sites by HIV protease [137].

Selberg et al. showed that activation of the METTL3/METTL14/WTAP RNA methyltransferase complex can promote m^6^A methylation in HIV-1 RNA to increase viral replication [138]. Therefore, in future studies, we can target m^6^A-related RMPs to reduce m^6^A in HIV-1 RNA and inhibit virus replication.

**Table 2 ijms-23-07571-t002:** Major roles of epitranscriptomic factors in HIV-1 infection.

Modification	Gene Symbol	Role in HIV-1 Infection	Refs
m^6^A	METTL3,METTL14	High expression promotes viral replication.	[136,138]
	YTDHF1-3	Controversial	[42,131,137]
m^1^A	TRMT6	necessary for virus replication	[139]
m^5^C	NSUN2	Mediates m^5^C of the HIV-1 RNA genome and facilitates viral protein synthesis.	[140]
	NOP2	Mediates m^5^C methylation of HIV-1 TAR and interferes with pre-transcription.	[141]
m^7^G	PIMT	Hypermethylation of m^7^G-cap by PIMT significantly promotes infectious virus production.	[142,143]
m^6^Am	PCIF1	Inhibit viral replication, limit HIV infection.	[144]
Ψ	Unknown	Unknown	
Nm	FTSJ3	Increase viral replication	[145]
ac^4^C	NAT10	Promote the expression of viral genes.	[146]

### 3.2. The Role of m^1^A Modification during HIV-1 Infection

Liquid chromatography-mass spectrometry (LC/MS) analysis of HIV-1 virion RNA combined with m^1^A profiling and deep sequencing methods revealed that HIV-1 virion RNA contains a large amount of 1-methyladenosine, but m^1^A is present in tRNA not present in the HIV-1 genome [147]. HIV-1 is a retrovirus, and the reverse transcriptase reversed by HIV-1 infection uses cellular tRNA as a substrate. The study by Hiroyuki Fukuda et al. elucidates the fundamental importance of tRNA m^1^A 58 modification in the early and late stages of HIV-1 replication and in cellular tRNA modification networks. TRMT61A and TRMT6 are responsible for m^1^A58 modification of cytoplasmic tRNAs. HIV-1 is a positive-strand RNA virus. HIV-1 is a positive-strand RNA virus. After entering the host cell, cytoplasmic tRNA3Lys is used as a primer, and then most of the viral RNA is degraded by RNaseH, leaving two short viral RNA’s polypurine tracts, which are used as primers for positive-strand DNA synthesis [148]. ‘Plus-strand strong-stop’ DNA is generated after the plus-strand enters the 3′ end of the tRNA 18nt terminated [139].After that, the plus-strand strong-stop DNA is transferred to the 3′ end of the template minus-strand DNA, and RT completes the synthesis of plus-strand DNA [149]. In experiments by Fukuda et al., they demonstrated that host cell tRNA m^1^A 58 is required for strong positive-strand blockade of retroviruses in vitro. In their study, integration of the HIV-1 genome into host cells was greatly reduced when the virus was generated from TRMT6 mutant cells. In the late stage of HIV-1 replication, the protein levels of HIV-1 capsid p24 and its precursors GagPol pr160 and Gag pr55 were reduced in TRMT6 mutant cell lines, and HIV-1 integrase and its precursor GagPol Pr160 and processing intermediates were greatly reduced, Tat protein decreased. Thus TRMT6 is required for the early steps of HIV-1 replication and for efficient late accumulation of HIV-1 protein and virus production [150]. Collectively, tRNA m^1^A58 modification plays an important role in the early and late stages of HIV-1 replication and in the tRNA modification network. tRNA m^1^A58 is required for reverse transcription to produce strong positive-strand termination in vitro.

### 3.3. The Role of m^5^C Modification during HIV-1 Infection

m^5^C promotes viral mRNA translation. In the study of Courtney, a highly prevalent modification of m^5^C was detected on HIV-1 gRNA in infected cells, with m^5^C levels about 14 times higher than cellular mRNA. In their study, they identified NSUN2 as the primary m^5^C “writer” on HIV-1 gRNA and showed that loss of NSUN2 reduced HIV-1 protein expression but did not affect viral RNA levels. Furthermore, NSUN2 deletion disrupts ribosome recruitment, affecting HIV-1 alternative splicing [140]. Kong et al. found that the m^5^C RNA methyltransferase NOP2/NSUN1 is an HIV-1 restriction factor. Proteomic studies have found that NOP2 protein is an RNA-binding protein associated with the 5′UTR of HIV-1 [151]. Through experiments, Kong et al. found that NOP2 blocks HIV-1 replication by inhibiting LTR/Tat-driven HIV-1 transcription, while also inhibiting HIV-1 transcription. Furthermore, NOP2 promotes HIV-1 latency, and its depletion favors HIV-1 reactivation under basal and latency-reversing agents (LRA) stimulation conditions. Mechanistically, NOP2 binds to the 5′ LTR of HIV-1, and then by binding and interacting with the TAR of HIV-1, this domain competes with the Tat protein of HIV-1 and promotes the m^5^C methylation of the TAR, blocking HIV-1 transcriptional elongation. The competition of NOP2 with Tat and binding to TAR are mediated by the catalytic domain (MTD) of RNA MTase [141]. Therefore, NSUN2 mediates m^5^C methylation of HIV-1 RNA genome, which is beneficial to viral protein synthesis, while NOP2 mediates m^5^C methylation of HIV-1 TAR and interferes with pre-transcription. Addition of m^5^C to HIV-1 RNA affects alternative splicing of its transcripts. HIV-1 has high levels of epitranscriptional m5C modifications, which in turn promote viral gene expression by regulating viral RNA splicing and promoting viral mRNA translation. The role of m^5^C in viral replication is an understudied area that deserves further attention.

### 3.4. The Role of m^7^G Modification during HIV-1 Infection

RNA polymerase II (RNAP II) can add a m^7^G cap to the 5′ end of nascent RNA [152]. The m^7^G cap helps initiate translation in mammalian cells, whereas uncapped pre-mRNA is degraded by 5′ exonuclease to accelerate decay [153,154]. While viral RNA capping is less studied, it is generally accepted that many viral RNAs, such as cellular RNAs, have a 5′ m^7^G cap [155]. There is evidence from in vitro studies that HIV-1 RNA is capped [156,157]. Yedavalli et al. showed by immunoprecipitation that HIV-1 transcripts do have m^7^G- or trimethylguanosine-caps (TMG-caps) to varying degrees. In experiments they found that the HIV-1 transcript initially transcribed was m^7^G-capped. Unspliced or partially spliced HIV-1 RANs containing RREs bind to Rev and then recruit PIMTs to these RNAs. In this way, PIMT is introduced into viral RNA, which then hypermethylates the m^7^G cap to TMG-CAPS. PIMT-mediated TMG-caps can promote RNA expression. In addition, experiments have also shown that PIMT can regulate the expression of Rev- and CRM-1–dependent Gag-Pol-RRE RNA, and TMG cap is a hallmark of the CRM-1 pathway [142]. What’s more, the study by Singh also demonstrated that the hypermethylation of HIV-1 m^7^G-cap significantly promotes the production of infectious virus. In addition, hypermethylation of viral mRNA leads to cap-binding complex exchange to heterodimeric CBP80/NCBP3, thereby expanding the functional capacity of HIV-1 in immune cells [143].

### 3.5. The Role of m^6^Am Modification during HIV Infection

The m^6^A methylation on 5′-terminal 2′-*O*-methylated A on mRNA is catalyzed by methyltransferase phosphorylation of CTD-interacting factor 1 (PCIF1) [61,62]. The study by Zhang et al. demonstrated that HIV reprograms host m^6^Am RNA methylome through viral Vpr protein-mediated PCIF1 degradation. In their study they found that HIV genomic RNA did not contain any m^6^Am modifications. In HIV infection, m^6^Am modification is reduced in many cells, which is mediated by Vpr-induced PCIF1 degradation. In addition, they found that PCIF1 is a broad HIV inhibitor that inhibits viral replication and limits HIV infection through methylation of ETS1 mRNA. The binding of ETS1 to the HIV promoter can reduce HIV transcription. However, PCIF1 does not methylate HIV genomic RNA. Briefly, PCIF1 inhibits HIV infection by enhancing the stability of the host m^6^Am gene, but HIV relieves this restriction by Vpr-induced PCIF1 degradation during pathogenesis, PCIF1 affects the replication phase of HIV [144]. PCIF1 may become a target for HIV-1 infection inhibition.

### 3.6. The Role of *Ψ* Modification during HIV Infection

7SK RNA is a highly conserved non-coding nuclear RNA in vertebrates [158]. Studies have shown that 7SK is a key regulator of P-TEFb dynamic homeostasis and activity, and P-TEFb is a specific host cofactor of HIV-1 transcription [159]. P-TEFb is part of the multi-subunit hyperelongation complex (SEC), which is recruited to viral LTR by HIV-1-encoded Tat protein to stimulate viral elongation [160,161]. Research by Zhao et al. shows that most of 7SK is pseudouridylated at position U250 by the predominant cellular pseudouridine synthase machinery. Pseudouridylation is key to the stabilization of 7SK snRNP because it mutates or is absent from DKC1 (the catalytic component of Box H/ACA RNP) at or near U250, thereby destroying 7SK snRNP and releasing P-TEFb to form superextension complex (SEC) and BRD4-P-TEFb complex. Tat then recruited SEC to the HIV-1 promoter to stimulate viral transcription and escape the incubation period [162]. Although this modification has not been found in viral RNA so far, the cell response to viral infection and its impact on viral infection can be revealed through the pseudo uridine reaction on host mRNAs.

### 3.7. The Role of Nm Modification during HIV Infection

In higher eukaryotes, the cellular innate immune system can distinguish endogenous mRNA from exogenous mRNA through the molecular signature of RNA 2′-*O*-methylation. When there is RNA without 2′-*O*-methylation modification in the cell, the cytoplasmic sensors Mda5 and RIG-I in the cytoplasm will recognize this type of exogenous RNA and further activate the expression of type I interferon (IFN) to exert antiviral function [163]. The genomes of some RNA viruses, such as flaviviruses, can encode their own viral 2′-*O*-MTase, whose encoded product can methylate adenine in the viral genome, evading the innate immune response of the immune system [164]. In their study, Ringeard et al. identified a DICER-independent TRBP complex containing FTSJ3 by purifying TAR RNA-binding protein (TRBP) and its interacting partners. The RNA-binding protein TRBP binds the 5′-terminal leader sequence TAR and Rev original of double-stranded RNA and promotes HIV expression. In vitro and in vivo experiments showed that FTSJ3 is a 2′-*O*-methyltransferase that is recruited to HIV RNA via TRBP. The transfection of HIV RNA expressed in FTSJ3 knockdown cells caused a significant increase in IFN-α/IFN-β compared with the control group. Later, the researchers used the knockdown of the RNA cytoplasmic receptors Mda5 and RIG-I cells to conduct experiments and proved that FTSJ3-TRBP completes immune escape through the Mda5 pathway through the 2′-*O*-methylation modification of HIV RNA by FTSJ3-TRBP. The FTSJ3-TRBP system can be hijacked to complete the 2′O-methylation modification of viral RNA to reduce the sensitivity of the MDA5-IFN pathway, reduce immune system recognition, and increase HIV-1 replication [145]. Furthermore, rRNA methyltransferase fibrillin (FBL) expression is directly controlled by p53, which inhibits FBL expression by directly binding to FBL, protecting the translation machinery and thus preventing aberrant rRNA methylation [165].

### 3.8. The Role of ac^4^C Modification during HIV Infection

The study by Tsai et al. mapped the ac^4^C residue on HIV-1 RNA to 11 distinct sites. Their studies identified that the cellular acetyltransferase NAT10 adds ac^4^C to HIV-1 transcripts at multiple discrete locations, which promote HIV-1 gene expression by increasing the stability of viral RNA while viral mRNA translation was not affected [146].

## 4. Conclusions

In the decades since RNA modifications were discovered, scientists have discovered multiple roles for dynamic RNA modifications. RNA modification can dynamically reprogram gene expression and regulate RNA stability, metabolism, splicing, translation, localization, transport, binding to other RNAs or binding to RNA-binding proteins. RNA modifications are also regulated by regulatory proteins. As the virus that causes AIDS, HIV-1 is also regulated by RNA modifications in the process of infecting the human body. Current studies have shown that several different RNA modifications can indirectly promote viral replication either by directly increasing viral RNA stability or translation, or by allowing the virus to evade innate immune recognition of viral transcripts. Therefore, the exploration of inhibitors that block viral RNA modification may inhibit the target of viral genes and promote the antiviral immune response of the host. In addition, this type of drug targets proteins associated with RNA modifications, so there is little chance of resistance or viral mutation. However, much remains unknown about the role of RNA modifications in HIV-1 infection. So, exploring how viruses modify, avoid or use RNA modification processes to regulate the expression of their own viral genes, and have a better understanding of the role of RNA modifications in HIV-1 infection may help in the search for new targets for antiviral therapy.

## Figures and Tables

**Figure 1 ijms-23-07571-f001:**
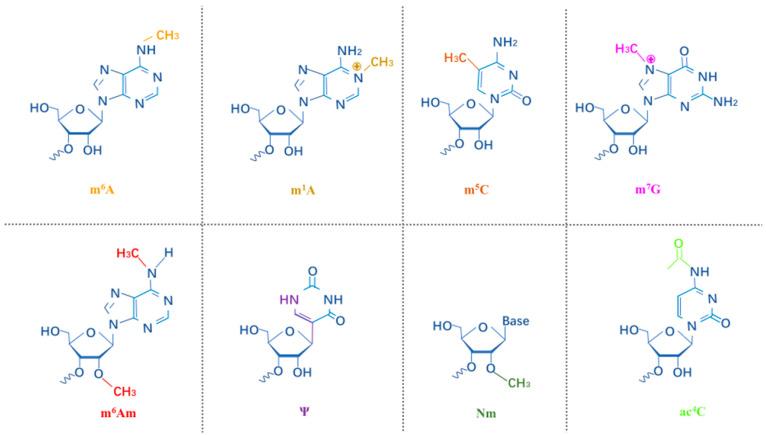
The structure of modifications in RNA. The indicated epitranscriptomic marks are m^6^A, m^1^A, m^5^C, m^7^G, m^6^Am, Ψ, Nm and ac^4^C.

**Figure 2 ijms-23-07571-f002:**
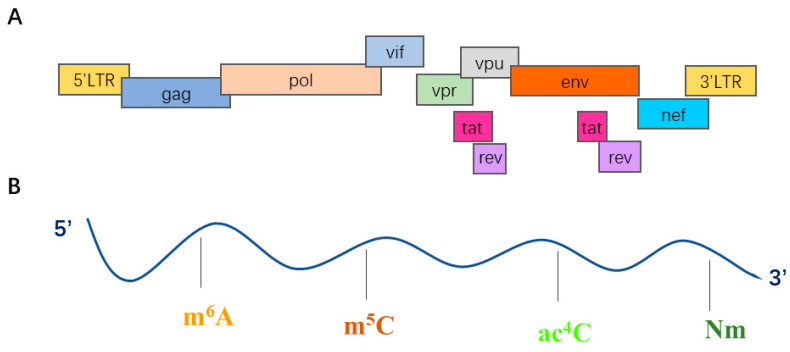
(**A**) Structure and organization of the HIV-1 genome. There are the reading frames of the genes coding for structural and regulatory proteins. (**B**) Epitranscriptomic modifications of HIV-1 RNA. The indicated epitranscriptomic marks are m^6^A, m^5^C, ac^4^C and Nm.

**Table 1 ijms-23-07571-t001:** Summary of writer and reader proteins for common mRNA modifications and the effects of these modifications on cellular RNA.

Modification	Writers	Readers	Roles on RNA	Refs
m^6^A	METTL3, METTL14, METTL16	YTHDF1-3, YTDHC1-2, HNRNPA2B1, HNRNPC, HNRNPG, IGF2BPs, eIF3, FMRP, LRPPRC	Splicing, Stability, Structure, Translation, Export	[19,23,25,33,39]
m^1^A	TRMT6/61A, TRMT61B, TRMT10C	YTHDF1-3, YTHDC1	Translation	[30,43,44,45,46,47,48,49,50,51]
m^5^C	NSUN, DNMT2, TRDMT	ALYREF, YBX1	Structure, Translation	[52,53,54,55,56]
m^7^G	METTL1/WDR4, Trm8/Trm82	Unknown	Splicing, Stability, Translation, Export	[57,58,59]
m^6^Am	PCIF1	Unknown	Regulating gene expression	[60,61,62]
Ψ	DCK1, PUS1-10	Unknown	Splicing, Stability,	[63,64,65,66,67,68]
Nm	FTSJ1	Unknown	Stability, Biogenesis, Metabolism, Function	[69,70,71,72,73,74,75,76,77]
ac^4^C	NAT10	Unknown	Stability, Translation,	[78,79,80,81,82]

## Data Availability

Not applicable.

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
