# Peer review of "The Role of RNA Modification in HIV-1 Infection"

_ijms, 2022, doi:10.3390/ijms23147571_

Round 1
Reviewer 1 Report
The manuscript by Wang et al reviews on the role of RNA modification in HIV-1 infection. They first introduced the varying types of RNA modifications, the corresponding readers, writers and erasers; then the roles of these modifications in HIV-1 infection were discussed.
Overall, this manuscript suffers from intensive English language issues, and needs careful proofreading.
More importantly, the review lacks a detailed description of the HIV-1 life cycle, which should be the basis of the discussion on RNA modification effects.
1. Figure 1. Authors should better describe the viral RNA genome components, annotate the varying RNA structures (e.g. TAR, etc) in the 5' and 3' UTRs, the different genes, and then map out in which regions the different RNA modifications are prevalent.
2. How do RNA modifications affect the HIV life cycle, e.g. in reverse transcription, replication and translation etc and what are the underlying mechanisms (e.g. shaping RNA structure dynamics, etc.)?
3. From what is known about these modifications, what can we learn in HIV-1 infection treatment? Can these lead to potential drug targets/design?
Without these aspects being discussed, the current manuscript only simply listed the research results from others and failed to synthesize from these results to novel contributions to the field.
Author Response
Dear Editor,
Thank you very much for your kind information regarding your comments and the reviewer’s criticisms for our manuscript entitled “The role of RNA modification in HIV-1 infection” . Those comments are all very valuable and very helpful revising and improving our paper, as well as the important guiding significance to our research. We have made a detailed reversion on the manuscript according to the editor’s criticisms and suggestions, and resubmit the revised manuscript for your consideration of publication in the Molecular Sciences, which we hope meet with approval.
Thank you again for your time and consideration.
The main corrections in the manuscript and the responds to the reviewer’s comments are as following:
The review lacks a detailed description of the HIV-1 life cycle, which should be the basis of the discussion on RNA modification effects.
Response: Thanks for your review. We have added dtailed description of the HIV-1 life cycle in the manuscript.
1. Figure 1. Authors should better describe the viral RNA genome components, annotate the varying RNA structures (e.g. TAR, etc) in the 5' and 3' UTRs, the different genes, and then map out in which regions the different RNA modifications are prevalent.
Response: Thanks to reviewer for your careful review, we have changed the Figure1 into Figure1 and 2. Figure1 is the structure of modifications in RNA. Figure 2 is structure and organization of the HIV-1 genome and epitranscriptomic modifications of HIV-1 RNA.
2. How do RNA modifications affect the HIV life cycle, e.g. in reverse transcription, replication and translation etc and what are the underlying mechanisms (e.g. shaping RNA structure dynamics, etc.)?
Response: Thanks to reviewer for your review. We have a more detailed explanation of how RNA modification affects the HIV cycle. But we have read less about the underlying mechanism of the impact, so there's less involved in the manuscript.
3. From what is known about these modifications, what can we learn in HIV-1 infection treatment? Can these lead to potential drug targets/design?
Response: Thanks to reviewer for your rigorous review. Several different RNA modifications can indirectly promote viral replication either by directly increasing viral RNA stability or translation, or by allowing the virus to evade innate immune recognition of viral transcripts. Therefore, the exploration of inhibitors that block viral RNA modification may inhibit the target of viral genes and promote the antiviral immune response of the host.
Reviewer 2 Report
RNA modifications are involved in the HIV-1 infection, so this type of review is necessary.
However, there are many careless mistakes in this manuscript. The author needs to rewrite the manuscript more carefully.
-p2 line2: ….2’-O-methylation (2 '-O-Me or Nm) and N4-acetylcytidine are introduced (ac4C) modification and its potential regulatory mechanism and role in HIV-1 infection.
->> I can’t understand the sentence.
-p4 line11: is the m1A position correct?? In human 28 rRNA, I think that the position of m1A is 1309. Please check it.
-P4 2.4 m7G section: m7G -containing tRNAs are primarily found in yeast and more recently in mice, …
>>In this sentence, it should be clearly stated that it is in eukaryotic tRNA.
-p4 line3:…usually located at position 9, 14, and 58..(43)
->> The reference is not suitable for the sentence.
-Where is Figure 2 ??
-References should be inserted in Table 1 and 2.
-Authors name: there is no name after “and”.
-Abstract :”h” of “human immunodeficiency virus” >>uppercase letter. I found some places the problem. Please check it more carefully.
-Introduction : Although “Post-transcriptional modification”is abbreviated as “PTM”, not required as it will not appear after that.
“WTAP” is also.
-Introduction : Since transfer RNA and ribosomal RNA are used, plural “s” is not required for the abbreviation.
-p2 section title :RNA modification>>RNA modifications
-Although the modified nucleosides are abbreviated, there are many places where it is written with the full name after that. Since it is duplicated, it should be shown by abbreviations.
The modified nucleosides in the tables can also be the abbreviations.
Although “pseudouridine(Ψ)” is shown on the line 3 of p2 line3, pseudouridine has already appeard in the Introduction. “(Ψ) “should be indicated in the introduction.
TET (p4. m5C section) is also.
-The “N” and “O” of “N6-methyladenosine”, “N6, 2’ -O-dimethyladenosine”, “5-Formyl-2’-O-methylcytidine” etc … are italic.
-There are many cases where there is space in places where space is not needed, and conversely there is no space!!
-There are many places where the comma is garbled.
-When quoting in the text, “last name et al.” , not “full name et al.”
-p4 line 12: Sylation?? Mistype?
-p4 line 13 : tRAN-like→tRNA-like
-p5 line 11 :”CTD” of the ”CTD-interacting factor 1” should be indicated as “C-terminal domain”.
-p6 line11 : E.coli>>italic, tRNA met >>tRNAMet
-P6 line25:NAT→NAT10
-p9 m7G section : premRNA→pre-mRNA, and “TMG” of the “TMG-caps” should be indicated as “trimethylguanosine”.
-p10 3.6 ψsection: There are “P-TEFb”と”PTEF-B”と”P-TeFb”. Are there any differences between them?
Pseudourine ylation →pseudouridylation?
-Figure 1 legend: Including ->>including
-Figure1: “2’O-methyl” of the m6Am is not colored.
Author Response
Dear Editor,
Thank you very much for your kind information regarding your comments and the reviewer’s criticisms for our manuscript entitled “The role of RNA modification in HIV-1 infection” . Those comments are all very valuable and very helpful revising and improving our paper, as well as the important guiding significance to our research. We have made a detailed reversion on the manuscript according to the editor’s criticisms and suggestions, and resubmit the revised manuscript for your consideration of publication in the Molecular Sciences, which we hope meet with approval.
Thank you again for your time and consideration.
The main corrections in the manuscript and the responds to the reviewer’s comments are as following:
- -p2 line2: ….2’-O-methylation (2 '-O-Me or Nm) and N4-acetylcytidine are introduced (ac4C) modification and its potential regulatory mechanism and role in HIV-1 infection.
->> I can’t understand the sentence.
Response: Thanks to reviewer for your careful review, we have corrected it. The sentence was changed into “In this paper, the m6A, m1A, m5C, m7G, m6Am, pseudouridine (ψ), 2'-O-methylation (2'-O-Me or Nm) and N4-acetylcytidine (ac4C) modifications (Fig.1) as well as their roles in HIV-1 infection are introduced.”
- -p4 line11: is the m1A position correct?? In human 28 rRNA, I think that the position of m1A is 1309. Please check it.
Response: Thanks for your careful review. We checked from the references, in human 28 rRNA, he position of m1A is 1309. We have corrected it.
- -P4 2.4 m7G section: m7G -containing tRNAs are primarily found in yeast and more recently in mice, …
>>In this sentence, it should be clearly stated that it is in eukaryotic tRNA.
Response: Thanks to reviewer for your rigorous review. We have made changes to emphasize in this sentence that in eukaryotic tRNA.
- -p4 line3:…usually located at position 9, 14, and 58..(43)
->> The reference is not suitable for the sentence.
Response: Thanks for your review. We have corrected it.
- -Where is Figure 2 ??
Response: Thanks to reviewer for your rigorous review. There is no Figure 2. We have corrected it.
- -References should be inserted in Table 1 and 2.
Response: Thanks to reviewer for your review, we have inserted the references in Table 1 and 2.
- -Authors name: there is no name after “and”.
Response: Thanks for your careful review. We have deleted the “and”.
- -Abstract :”h” of “human immunodeficiency virus” >>uppercase letter. I found some places the problem. Please check it more carefully.
Response: Thanks for your review. We have corrected the error here and elsewhere in the manuscript
- -Introduction : Although “Post-transcriptional modification”is abbreviated as “PTM”, not required as it will not appear after that.
“WTAP” is also.
Response: Thanks to reviewer for your review, we have removed it.
- -Introduction : Since transfer RNA and ribosomal RNA are used, plural “s” is not required for the abbreviation.
Response: Thanks for your review. We have corrected it.
- -p2 section title :RNA modification>>RNA modifications
Response: Thanks for your careful review. We have corrected it.
- -Although the modified nucleosides are abbreviated, there are many places where it is written with the full name after that. Since it is duplicated, it should be shown by abbreviations.
The modified nucleosides in the tables can also be the abbreviations.
Although “pseudouridine(Ψ)” is shown on the line 3 of p2 line3, pseudouridine has already appeard in the Introduction. “(Ψ) “should be indicated in the introduction.
TET (p4. m5C section) is also.
Response: Thanks for your review. We have checked and corrected it.
- -The “N” and “O” of “N6-methyladenosine”, “N6, 2’ -O-dimethyladenosine”, “5-Formyl-2’-O-methylcytidine” etc … are italic.
Response: Thanks for your careful review. We have corrected it and made them italic.
- -There are many cases where there is space in places where space is not needed, and conversely there is no space!!
Response: Thanks for your careful review. We have checked and corrected it.
- -There are many places where the comma is garbled.
Response: Thanks for your review. We have checked and corrected it.
- -When quoting in the text, “last name et al.” , not “full name et al.”
Response: Thanks for your careful review. We have checked and changed the name quoted in the manuscript to last name
- -p4 line 12: Sylation?? Mistype?
Response: Thanks for your review. We have checked and corrected it. It is a mistake.
- -p4 line 13 : tRAN-like→tRNA-like
Response: Thanks for your careful review. We have corrected it.
- -p5 line 11 :”CTD” of the ”CTD-interacting factor 1” should be indicated as “C-terminal domain”.
Response: Thanks for your review. We have corrected it.
- -p6 line11 : E.coli>>italic, tRNA met >>tRNAMet
Response: Thanks for your careful review. We have corrected it.
- -P6 line25:NAT→NAT10
Response: Thanks for your review. We have corrected it.
- -p9 m7G section : premRNA→pre-mRNA, and “TMG” of the “TMG-caps” should be indicated as “trimethylguanosine”.
Response: Thanks for your careful review. We have corrected it.
- -p10 3.6 ψsection: There are “P-TEFb”と”PTEF-B”と”P-TeFb”. Are there any differences between them?
Pseudourine ylation →pseudouridylation?
Response: Thanks for your careful review. These three worlds are all P-TEFb. We have checked and corrected it.
- -Figure 1 legend: Including ->>including
Response: Thanks for your careful review, We have corrected it.
- -Figure1: “2’O-methyl” of the m6Am is not colored.
Response: Thanks for your review. We have checked and colored it.
In all, the comments from editor and reviewers are quite helpful to our manuscript, and we have revised our manuscript point by point. Once again, thank you very much for your good comments and suggestions.
Best regards,
Shuqi Wang
Beijing Key Laboratory for Animal Genetic Improvement, National Engineering Laboratory for Animal Breeding, Key Laboratory of Animal Genetics and Breeding of the Ministry of Agriculture, College of Animal Science and Technology, China Agricultural University.
Round 2
Reviewer 1 Report
The authors have adequately addressed my points. I only suggest to add two key references to support the roles of m6A and Nm modifications in translation (dynamics): PMID: 26751643 (https://pubmed.ncbi.nlm.nih.gov/26751643/) and PMID: 29459784 (https://pubmed.ncbi.nlm.nih.gov/29459784/).
Author Response
Dear Editor,
Thank you for your decision and constructive comments on my manuscript entitled “The role of RNA modification in HIV-1 infection”. We have carefully considered the suggestion of Reviewer and add two key references to support our views. We have tried our best to improve and made some changes in the manuscript. [15], [107]
Thank you again for your time and consideration.
In all, the comments from editor and reviewers are quite helpful to our manuscript, and we have revised our manuscript point by point. Once again, thank you very much for your good comments and suggestions.
Best regards,
Shu-qi Wang
Beijing Key Laboratory for Animal Genetic Improvement, National Engineering Laboratory for Animal Breeding, Key Laboratory of Animal Genetics and Breeding of the Ministry of Agriculture, College of Animal Science and Technology, China Agricultural University.